# Paving the Way to Solid Tumors: Challenges and Strategies for Adoptively Transferred Transgenic T Cells in the Tumor Microenvironment

**DOI:** 10.3390/cancers14174192

**Published:** 2022-08-29

**Authors:** Franziska Füchsl, Angela M. Krackhardt

**Affiliations:** 1Klinik und Poliklinik für Innere Medizin III, School of Medicine, Technische Universität München, Klinikum rechts der Isar, Ismaningerstr. 22, 81675 Munich, Germany; 2German Cancer Consortium of Translational Cancer Research (DKTK) and German Cancer Research Center (DKFZ), 69120 Heidelberg, Germany; 3Center for Translational Cancer Research (TranslaTUM), School of Medicine, Technical University of Munich, 81675 Munich, Germany

**Keywords:** adoptive T cell transfer, transgenic T cells, TCR, CAR, solid tumors, tumor microenvironment, lymphocyte engineering

## Abstract

**Simple Summary:**

Adoptive transfer of T cells targeting tumors still remains mainly ineffective in solid entities. In this review we discuss challenges related to the tumor microenvironment (TME) and promising strategies to improve tumor control by engineering the TME or the transferred T cells themselves.

**Abstract:**

T cells are important players in the antitumor immune response. Over the past few years, the adoptive transfer of genetically modified, autologous T cells—specifically redirected toward the tumor by expressing either a T cell receptor (TCR) or a chimeric antigen receptor (CAR)—has been adopted for use in the clinic. At the moment, the therapeutic application of CD19- and, increasingly, BCMA-targeting-engineered CAR-T cells have been approved and have yielded partly impressive results in hematologic malignancies. However, employing transgenic T cells for the treatment of solid tumors remains more troublesome, and numerous hurdles within the highly immunosuppressive tumor microenvironment (TME) need to be overcome to achieve tumor control. In this review, we focused on the challenges that these therapies must face on three different levels: infiltrating the tumor, exerting efficient antitumor activity, and overcoming T cell exhaustion and dysfunction. We aimed to discuss different options to pave the way for potent transgenic T cell-mediated tumor rejection by engineering either the TME or the transgenic T cell itself, which responds to the environment.

## 1. T Cells: Essential Players across Immunotherapeutic Approaches

Various immunotherapeutic approaches targeting cancer focus on T cells. CD8^+^ T cell responses appear to play a major role in the success of immune checkpoint inhibitors (ICI) such as anti-PD-1/-PD-L1, anti-CTLA4, and others [1,2,3,4]. Bi- and tri-specific antibodies are used to attract T cells to the tumor site (the bispecific T cell engager (BiTE) platform) [5,6,7]. CD19- and BCMA-targeting chimeric antigen receptor (CAR)-transgenic T cell products show effective and safe clinical responses [8,9,10,11]. Eventually, the endogenous TCR repertoire is employed for highly personalized antitumor treatment by tumor-infiltrating lymphocyte (TIL) or T cell receptor (TCR)-transgenic T cell infusions [12,13,14,15]. Overall, numerous applications suggest the potential of T cells for antitumor therapy.

### 1.1. T Cells as Key Players of the Antitumor Immune Response

A complex interplay between a variety of different immune cell populations is indispensable for effective tumor control. Yet, T cells, key players in the adaptive immune system, play an essential role in the antitumor response. The increased infiltration of tumors with CD8^+^ T cells is largely correlated with prolonged survival across different entities [16,17,18,19,20,21,22]. CD8^+^ TCRαβ T cells exert cytolytic functions in antiviral and antitumor defense [22] and produce large amounts of cytotoxic molecules such as interferon-γ (IFN-γ), tumor necrosis factor-α (TNFα), perforin, and granzymes. For this reason, they are also known as cytotoxic T cells (CTLs) [23]. 

The quantity of cytotoxic cells alone does not guarantee potent antitumor responses. CD8^+^ T cells do not act in isolation; CD4^+^ TCRαβ T cells, for example, play a well-established role as T helper cells (T_H_), interacting with professional antigen-presenting cells (APCs), promoting CD8^+^ T cell-priming and providing cytokines for effective T cell responses [24]. 

Devoted to selectively suppressing T effector cells and maintaining self-tolerance, the T cell pool also contains immunosuppressive regulatory T cells (T_reg_ cells). Conventionally, T_reg_ cells express the transcription factor FOXP3 and the IL-2Rα chain (CD25) (thymus-derived tT_reg_ or peripherally induced pT_reg_, e.g., Th3 cells), but there are also FOXP3-independent, peripherally induced subsets (e.g., Tr1 cells) [25,26,27]. Suppressing immune functions, T_regs_ are often involved in hampering antitumor immune processes in the tumor microenvironment (TME) [28].

### 1.2. Targets for T Cell-Based Immunotherapeutic Approaches

The quality of a T cell response—for CD8^+^ or CD4^+^ cells—is determined by the recognition of a specific target antigen presented on a major histocompatibility complex (MHC) on the surface of the tumor cell via the cognate TCR. Thus, the TCR is the cell’s main trigger for antigen-specific clonal expansion and cytokine secretion. It consists of disulfide-bonded α- and β-chains composed of a constant and a variable domain, of which the latter contains the antigen-binding site and undergoes somatic recombination on the genetic level during maturation (the so-called V(D)J-recombination of the variable, diversity, and joining the gene sequence) [29]. This process results in more than 10^13^ stochastically possible TCR sequences for each human being [30,31] of which usually only the non-self-reactive survive thymic selection as a means of central tolerance, preventing autoimmunity. [32] TCR targets are peptides that have undergone intracellular processing and presentation machinery, ending up on the binding cleft of MHC-class I or II complexes—the first presenting these antigens to CD8^+^, the latter to CD4^+^ T cells [33]. The initial contact between a (at this stage) naïve T cell and its respective antigen, so-called priming, mostly takes place in secondary lymphoid tissues, where professional APCs (dendritic cells (DCs), macrophages, or B cells) with important ligands for costimulatory receptors on the T cell-surface have processed these antigens. Subsequently, clonal expansion is initiated and tumor-specific T cells start to infiltrate the tumor site [34].

Rosenberg and colleagues were the first to demonstrate that collecting TILs, at first from mice and later from melanoma patients, and then expanding them in vitro and reinfusing these cells back into the patient provoked potent antitumor responses. This strongly suggested the presence of tumor-reactive TCR clonotypes in the tumor tissue, but also resulted in the question about their target [35,36,37]. When it comes to T cell therapies, TCR- and CAR-based target identification might be the most challenging bottleneck. In principle, the following options exist: over- or selectively expressed self-antigens in the adult, differentiation, and cancer-testis antigens (CTA) or neoantigens. 

Over the past several years, self-antigens, regularly expressed in the adult body with defined tissue-restricted expression, reached a prominent role in tackling hematological malignancies via CARs. Regarding hematological cancers, lineage markers such as CD19 for B cell lymphoma [8,9,10] or BCMA for multiple myeloma (MM) [11,38,39] can be targeted at the expense of also eliminating the respective healthy cell population. B cells, for example, are dispensable, up to a certain point (at least temporarily) [40]. However, since compared to pathogens tumor cells are not foreign, per se, the sum of the peptides presented by MHC class I or II molecules, the so-called immunopeptidome, contains mainly self-peptides. Due to the mechanisms of central tolerance they are, in most cases, suboptimal targets for endogenous TCRs [41]. For solid tumors, no distinct surface-expressed lineage markers exist, and targeting self-antigens comes at the risk of on-target off-tumor toxicity. Therefore, target choice and therapy safety assessments become crucially important. In melanoma, melanocyte-specific proteins, such as epitopes of glycoprotein 100 (gp100), serve as target candidates, and this is exemplarily so for the recently approved BiTE Tebentafusp [42]. Another melanoma-associated antigen, PRAME (Preferentially expressed Antigen in Melanoma), has been detected in medulloblastoma and synovial sarcoma, promising a potential therapeutic target [43,44]. A third example, the tumor-associated antigen (TAA) mesothelin (MSLN), is widely expressed in malignant pleural mesotheliomas (MPM), pancreatic, ovarian, and some lung cancer entities [45,46]. However, it can also be detected at low levels on peritoneal, pleural, and pericardial mesothelial surfaces [47]. MSLN-targeted CAR-T cells show manageable side effects and reasonable safety profiles [48], yet possess only limited clinical efficacy compared with the response rates of CARs in hematological cancers [49,50]. Similarly, TCRs targeting MSLN showed antitumor activity in a murine model yet became progressively dysfunctional over time. [51] Other self-antigens, however, are more specific for the tumor tissue. L1CAM (CD171), for example, is expressed in a supposedly glycosylated form by neuroblastoma, and an antibody recognizing this glycosylation-dependent epitope can be used for the generation of a CAR-product [52].

Secondly, developmental antigens re-expressed in the tumor tissue display promising candidates for sufficient tumor specificity and, therefore, a reduced risk for off-tumor toxicity. The oncofetal protein Claudin 6 (CLDN6), virtually absent in normal body tissues, is aberrantly—but frequently—expressed in various cancer entities and therefore targeted by CAR-approaches [53,54]. Furthermore, so-called cancer-testis antigens (CTAs), only found in gametogenic tissue in the adult body otherwise, are reactivated during tumorigenesis. Many of the TCRs already assessed in clinical studies target some of the most abundantly expressed immunogenic CTAs such as NY-ESO-1 or members of the MAGE-family [55,56]. 

Nevertheless, reactivities against both kinds of antigen cannot be solely responsible for endogenous antitumor activity, since self-reactive TCRs should be mostly eliminated by central tolerance; additionally, those remaining in the repertoire should be expected to show a low avidity [57,58]. Since a tumor deviates from normal body cells due to mutations during tumorigenesis, a third class of antigens emerges. To date, it is assumed that somatic mutations in the tumor genome, as well as aberrant posttranslational modifications, can induce aberrant peptide ligands differing from the self. These can be recognized by TCRs upon their presentation on HLA-molecules. These mutated peptide ligands, termed neoantigens, promise highly tumor-specific—and therefore safe and extremely personalized—tumor targeting [59].

## 2. T Cell-Based Adoptive Cellular Therapy for Solid Tumors: Where Are We in the Clinic?

### 2.1. TCR-Based Approaches

Reinfusing TILs, as pioneered by Rosenberg et. Al., was the historical predecessor of adoptive cellular therapy (ACT) regimens for tumor treatment and led to impressive remissions in metastasized melanoma [36]. By then, rather than TCR sequences, only tumor-reactive T cell clones were isolated, which can—despite general technical feasibility for many solid cancer entities—be limiting due to their low TIL infiltration, lack of immunogenicity, or reduced T cell fitness after several rounds of chemotherapeutics. Attempts to increase the percentage of tumor-reactive T cells via the selection of surface activation markers, e.g., CD137+, CD134+ or PD-1+ cells, have been made but have remained laborious and time-consuming [60,61,62]. Recently, single-cell genome- and transcriptome-sequencing methods have gained increasing influence in T cell-identification pipelines [63,64,65].

Meanwhile, the idea of transferring TCR genes, and thereby TCR specificity, from one T cell onto another was published in 1986 [66]. This became the direct ancestor of the concept of identifying tumor-reactive TCRs against TAAs in one patient and transferring these receptors off-the-shelf to other HLA-matching tumor patients with the expression of the identical antigen. In 2004, the first trial with 15 refractory melanoma patients was launched with a melanoma antigen recognized by T cells 1 (MART1)-specific HLA-A02-restricted TCR. Two patients showed a complete clinical regression of disease after the transfer of autologous, genetically engineered T cells [12]. Four years later, the first clinical TCR-T cell trial in another solid tumor entity, a metastatic synovial sarcoma, was launched with NY-ESO-1-reactive TCRs. Four of the six sarcoma patients responded to the treatment [55]. Meanwhile, many studies on TCR-based T cells in solid tumors have been launched (see Table 1), continually demonstrating the potential of these cellular therapies in the future, but also highlighting the challenges ahead. 

### 2.2. CAR-Based Approaches

Despite TCRs predating tumor therapy, CARs drew ahead in the race between both ACT approaches. While TCR-T cell products have thus far remained in clinical studies, several CAR products have been FDA- and EMA-approved for the treatment of hematological malignancies, with impressive clinical response rates. Among these are the CD19-targeting CAR constructs approved for the treatment of B cell lymphoma: axicabtagene ciloleucel (Axi-cel) [8], tisagenleucel (Tisa-cel) [10,75,76], and lisocabtagene maraleucel (Liso-cel) [77]—the first of these with CD28 co-stimulation; the latter two with 41BB co-stimulation. Idecaptagene vicleucel (Ide-cel) was approved for treating multiple myeloma (MM) by targeting B cell maturation antigen (BCMA) and including 41BB-costimulation [11,38]. 

CARs are synthetically engineered receptors mimicking TCR signaling, according to our understanding of T cell activation. In contrast to TCRs, CARs do not recognize peptides bound to MHC-complexes. Instead, they comprise single-chain variable fragments (scFv) from antibodies targeting an epitope of a surface-expressed protein [78]. This already suggests one main difference between the two approaches: while the target-repertoire of CARs is limited to surface antigens—only about 20–30% of all encoded proteins are expressed on the cell’s surface [79] and only a fraction of these is accessible for antibody binding—HLA-presented peptides also cover the intracellular proteome, and therefore grant a much broader target repertoire for TCRs. In CARs, the scFv differ from TCRs by being directly linked to an intracellular CD3ζ-chain containing the necessary binding sites (immunoreceptor tyrosine-based activation motifs, ITAMs), provoking T cell activation upon TCR ligation. [80,81] Since this first generation of CAR-constructs, which contained a CD3ζ (or CD3γ) chain alone [82,83,84], costimulatory elements—mostly the intracellular domain of CD28 or 41BB—were added to magnify the signaling effects and induce the more efficient killing of tumor cells [78]. These CARs of the second generation are the ones currently in use for clinical applications. Further engineering, however, has already led to a third and fourth generation with either two costimulatory domains or additional modifications such as off-switches, inducible cytokine secretion, or chemokine receptor expression [78,85]. Most of these constructs, however, have not yet been proven to be superior in clinical assessments.

Notwithstanding the success of CARs in hematological cancers, the treatment of solid tumors has remained rather ineffective to date [86]. The major obstacles and potential engineering strategies for both ACT approaches in the TME are outlined in the following section.

## 3. Challenges for T Cells in the Tumor Microenvironment

T cells encounter multifaceted obstacles in the microenvironment of a solid tumor (Figure 1). In principle, the therapeutic concept of TIL-therapy eradicating tumor tissue provides, on the one hand, evidence for the presence of immunogenic tumor antigens as well as preexisting tumor-reactive T cell clones in the patient’s TCR repertoire and even intratumorally. On the other hand, it raises the question as to why the patient’s immune system obviously does not efficiently control the tumor. 

### 3.1. Infiltrating the Tumor: Cold and Hot Tumor Microenvironment

Firstly, lacking an efficient homing mechanism to the tumor site, no tumor-specific clonotype can unleash antitumor activity. The identification of T cell infiltration into the tumor as one major predictive factor for survival led to the classification of outcomes in colorectal carcinoma based on an immune score rather than tumor staging [19,87,88]. It also resulted in the introduction of the now commonly used terms hot (inflamed) and cold (non-inflamed) tumors to describe the level of immune infiltration [89,90]. Primarily, the immune type of tumors range from inflamed (hot), immunosuppressed, and excluded to cold, with gradually lower numbers of T cells present in the tumor (see Figure 1). Those few T cells in the immunosuppressed tumor bed lack functional capacity due to immunosuppressive signaling. In the excluded state, immune cells accumulate at the outer tumor border, the so-called invasive margin, without further invasion. Cold tumors are not infiltrated by T cells at all [91,92]. This demonstrates the physical, immunological, and metabolic barriers impeding immune tumor control via subverting T cell trafficking and the penetration of the tumor parenchyma.

#### 3.1.1. Targeting Tumor Vessels

The first step toward effective cellular-based tumor therapies is T cells’ egress from the blood vessel at the tumor margin. In the hypoxic TME, the continued production of pro-angiogenic factors by tumor cells (e.g., vascular endothelial growth factor (VEGF)) stands in disbalance with antiangiogenic factors. Aberrant blood vessel formation is promoted. [93] Cancer cells sprout new vessels via angiogenesis, recruit endothelial cells from the bone marrow, or hijack host vessels. Such rapidly growing and reorganizing vessels show structural malformations, a leaky endothelium, and an abnormal, heterogeneous blood flow, not only aggravating the supply of nutrients but also chemotherapeutic agents [94]. Moreover, T cell arrest and transmigration require activated integrins on the endothelium of the target tissue. The lower expression of these adhesion molecules, such as vascular cell adhesion molecule-1 (VCMA1) or intercellular adhesion molecule-1 (ICAM1), on the irregular tumor vessel endothelium deteriorates the extravasation of immune cells and is termed endothelial anergy [95,96,97,98]. 

It has been described that the irregular endothelium compromises immune penetration for endogenous as well as adoptively transferred T cells, resulting in non-inflamed, cold tumors. One approach to increasing T cell infiltration is the administration of angiogenesis inhibitors, most of them targeting VEGF. Primarily, these inhibitors deprive the tumor of a blood supply to starve cancer cells of nutrients and oxygen and directly kill them. However, simultaneously, a reduced tumor perfusion is accompanied by lower T cell infiltration—at first view, the opposite of a strategy to switch a cold tumor into a hot tumor. However, by choosing lower, vascular-normalizing (rather than antiangiogenic) doses of VEGF-inhibitors, a synergistic effect of anti-VEGF and immunotherapy has been reported [99]. In part, this might be explained by an amelioration of the permeability of the tumor vessels for T cells and decreased interstitial pressure in a narrow time window and dose range. Furthermore, VEGF directly interacts with various immune cells, amongst them T cells, macrophages, dendritic cells (DCs), and myeloid-derived suppressor cells (MDSCs), as reviewed elsewhere in more detail [100,101]. All these interactions are additionally influenced by the VEGF-blockade and potentially contribute to antitumor effects. This example illustrates the fine-tuned interplay in the TME based on single growth factors. Apart from VEGF, other molecules were identified for the stabilization of the intratumoral vascular network, partly mediated by pericytes, such as low doses of TNFα or LIGHT [102,103,104].

In some models, however, only a combinatorial treatment with low doses of antiangiogenic therapy and the inhibition of programmed-death receptor 1 or its ligand (PD-1/PD-L1) improved survival. Apart from vascular normalization, antiangiogenic treatment was shown to induce PD-L1 expression in the TME of relapsing tumors as well as PD-1 on the surface of T cells. Abrogating such adaptive immunosuppressive effects via simultaneous ICI therapy sustained the therapy response. The formation of high endothelial venules (HEVs) with an activated endothelium necessary for the T cells’ egress was induced and thus T cell infiltration and activation were promoted. Thus, both treatments may reinforce each other synergistically [105,106,107]. Further clinical trials have already been launched for the combination of PD-L1-blockade and VEGF-inhibitors [108] as well as PD1-blockade and VEGF-receptor inhibitors [109]. These examples for combination therapy vividly demonstrate the sensitive balance between anti- and pro-tumorigenic factors and illustrates the need for a better understanding and additional treatment for the effective infiltration of either endogenous or adoptively transferred T cells into tumors.

#### 3.1.2. Application of Proinflammatory Stimuli

Converting cold tumors into hot tumors can also be attempted by less specific proinflammatory stimuli. At low doses, the proinflammatory effect of local irradiation can remodel the anergic endothelium to re-express activated integrins and selectins, which might increase T cell extravasation [110,111,112]. Moreover, radiotherapy-mediated damage leads to immunogenic cell death activating the immune system. This becomes especially evident as the so-called abscopal effect, where the local irradiation of a tumor leads to the shrinkage of distant tumors outside the radiation field [113]. Higher radiation doses, however, damage the endothelium, deteriorate immune infiltration, and necessitate judicious balancing. Further local manipulations, such as radiofrequency ablation therapy, photothermal therapy, or high-intensity focused ultrasound, can also help to increase T cell infiltration by triggering local inflammatory processes [114,115,116]. Further stimulatory treatments, such as the exemplary intravesical Bacillus Calmette-Guérin (BCG)-treatment in urothelial malignancies, also partly rely on such local immune activation by immunogenic cell death [117,118].

While it was previously thought that chemotherapy exerted its antitumor effects primarily by directly killing tumor cells and inhibiting their proliferation, it is now believed that tumor control also relies on immune stimulation. As for radiation therapy, immunogenic cell death after chemotherapy was shown to elevate immune activation. In a murine model, CAR-T cell infiltration and sensitivity to anti-PD-L1 therapy both increased after chemotherapy [119]. 

In any case, local tissue damage was accompanied by an increased release of TAAs, their uptake by DCs and macrophages, and a subsequent T cell priming, similarly to the concept of cancer vaccination. Moreover, mediated by proinflammatory cytokines such as IL-1β, TNF-α, or type I and II interferons (IFN) [120], chemokine secretion and upregulated adhesion molecules at the inflamed tumor site resulted in T cells’ ameliorated homing and infiltration [112,121,122]. 

In addition, DNA from damaged tumor cells led to the activation of the cyclic GMP-AMP (cGAS)-dependent and the stimulator of IFN genes (STING)-dependent pathways, resulting in a type I-IFN-release from DCs [123]. An intact cGAS-STING cascade in adoptively transferred CD8^+^ T cells was proven essential for the potency of antitumor responses in mice, maintaining stemness by the regulation of the transcription factor TCF1 and the restraint of AKT-signaling [124]. In preclinical mouse models, STING-agonists such as DMXAA were shown to be effective for tumor regression by triggering the cooperation between T cells and myeloid cells [125], thereby causing them to enhance CAR-T cell trafficking to the tumor site [126]. Due to the structural divergence in key amino acid residues of STING between humans and mice, the clinical translation required the identification and development of further STING-agonists. While synthetic cyclic dinucleotides were developed and are currently being tested for intratumoral administration [127,128], many approaches are being pursued to efficiently activate STING as an immune adjuvant for the immunomodulation of tumor environments [129].

Overall, radio-, chemotherapy, and immunomodulatory agents bear a potential for use in combination strategies with ACT by inducing local inflammatory stimuli. Paradoxically, inflammation also entails immunosuppression, most likely counterregulatory mechanisms, such as the upregulation of PD-L1 on tumor cells, immune cells, and the endothelium, which additionally requires combinatorial regimens of ACT with checkpoint inhibitors [130].

#### 3.1.3. Engineering T Cells to Alter the Cytokine Milieu

Synergistically, some groups have engineered not only the TME but the T cells themselves by radiotherapy. Splenic irradiation in mice upregulated chemokine receptor expression on T cells, which in turn promoted tumor infiltration [131]. Despite the questionable feasibility and toxicity for T cells and surrounding tissue in human patients, such attempts demonstrated that increasing immune cell infiltration into the TME was not solely reached by engineering the TME alone. It might also be achieved by modulating and engineering the T cells themselves. 

To date, no universal chemokine–receptor axis suitable for a broad variety of solid entities has been identified due to the diverse chemokine profiles of different tumors [132]. For lymphocyte extravasation, chemokine receptors, such as CXCR3 and CCR5, expressed on effector T cells are of central importance. Their corresponding ligands, CXCL9, CXCL10, CCL3, CCL4, and CCL5, are upregulated in inflamed tissue in an IFN-dependent manner [133,134]. Therefore, it is reasonable that through the gene transfer of different chemokine receptors, such as CXCR2, CCR2, CXCR6, or CCR8 in TCR- or CAR-T cells, the infiltration at the tumor site could be elevated in preclinical models [135,136,137,138,139]. For example, CXCR3 was identified to be of special importance for CD8^+^ T cells’ homing ability [140] and chemotherapeutically increased secretion of its ligands (CXCL9, CXCL10, and CXCL11) enhanced intratumoral T cell accumulation [141]. These findings showed CXCR3 to be a promising candidate for further T cell engineering in ACT. 

Along this line, a newly designed third generation of so-called armored CARs comprises so-called TRUCKs (T-cells Redirected towards Universal Cytokine Killing). CAR-T cells that can home to the tumor have been employed to modulate the cytokine milieu in the TME by creating an immune-activating positive feedback loop. Antitumor responses and T cell expansion were augmented via either the constitutive or—to avoid systemic toxicity—inducible secretion of, for instance, IL-12 [142,143,144,145], IL-18 [146], or IL-15 [147,148]. In that manner, CARs may become vehicles to switch the TME from an immunosuppressive to a pro-inflammatory state in addition to their direct cytotoxic capacity. 

Similarly, it is also possible to alter the effects of specific cytokines on CAR-T cells by manipulating their response to these molecules. For example, hybrid receptors combining elements of IL-7- and IL-2-receptors were used to convert the signal of IL-7-binding into the intracellular proliferative stimulus of IL-2 signaling cascades by simultaneously circumventing the IL-2-induced formation of immunosuppressive regulatory T cell populations [149]. Immunosuppressive signals from the TME, similar to TGF-β, were attenuated by introducing a dominant negative receptor (dnTGF-βRII) into CAR-T cells to intercept TGF-β without inhibitory downstream signaling [139,150]. This was already tested in a phase I clinical trial [151]. Moreover, costimulatory receptors or ligands, such as CD40L, can be constitutively expressed on transgenic T cells augmenting tumor responses by engaging with their counterpart [152]. Such approaches, however, must prove their clear superiority in the patient compared to the second-generation CAR-constructs currently used in clinics.

Third, the forced local secretion of chemokines by the transgenic CAR-T cells themselves is another strategy for attracting T cells to the tumor site in a positive-feedback loop. CCL19- and CCL21-secreting CARs were reported to increase T cell infiltration into the tumor [153,154,155]. Additionally, in part, the first clinical results for CCL19-incorporation combined with IL-7 overexpression in CAR-T cells show an improved antitumor capacity [153].

### 3.2. Effectively Targeting and Eradicating Tumor Cells

#### 3.2.1. TCR-T Cells and Peptide-MHC-Complexes

##### Target Identification

Once having left the blood vessel and reached the tumor site, the highly specific T cells can only act upon the recognition of their cognate antigen—as outlined previously, a TAA, CTA, or neoantigen. For TCR-based approaches, currently, the identification of suitable target peptides presented on HLA-molecules remains a major challenge for the broader application of ACT in solid tumors; unlike many hematological malignancies affecting one cell lineage with distinct surface markers, there is no particular antigen overexpressed for solid entities to be identified. This is underlined by the different strategies that some of the largest TCR-T cell companies follow [156] (see also Table 1). 

Independent of the kind of antigen, all TCR targets must be presented on an HLA-molecule on the tumor cell surface to make the tumor detectable for T cells. Differences in the individual tumor clones’ proteomes, and subsequently their immunopeptidomes, develop during tumorigenesis. The loss of DNA integrity due to DNA damage and inefficient repair mechanisms leads to the acquisition and accumulation of somatic mutations over time, resulting in a metabolic and proliferative potential to evolve into a tumor clone. Within the adapting tumor and immune environment, natural selective pressure selects for further subclones from the original combination of mutations [157,158,159]. 

For over-expressed self-antigens and CTAs, their detection within the tumor tissue remains relatively simple, as it is based on methods such as immunohistochemistry (IHC), PCR, western blot, and targeted or whole exome sequencing (WES). For example, New York Esophageal Squamous Cell Carcinoma-1 (NY-ESO-1), first described in a patient with an esophageal carcinoma in 1997 [160], was found to be expressed in various other different solid cancer entities, e.g., metastatic melanoma, synovial sarcoma, bladder cancer, head and neck cancer, ovarian cancer, or medulloblastoma [161,162,163,164,165,166]. Multiple immunotherapeutic approaches—TCR- and vaccine-based—have been launched since then, and epitopes of NY-ESO-1 are considered some of the most immunogenic CTA-derived peptides [167]. However, the actual presentation of antigen-derived peptides on the tumor cell surface in the individual patient cannot be distinguished by all the methods stated. These days, algorithms based on the individual HLA-type of the patient predict which peptides are most likely presented on the cell surface upon CTA expression [168,169]. However, expression analyses thus far barely dissolve the single cell-level and therefore lack information about CTA-peptide-presentation-negative tumor fractions. Since immunoediting due to therapeutic or immunogenic selection pressure favors antigen-negative tumor clones, those might remain unrecognizable for cognate TCRs, leading to refractory and relapsed disease stages. The further development of methods for quantification of surface-expressed peptide-MHC-complexes will thus be necessary to better understand the impact of molecular antigen presentation and density on T cell functionality [170,171].

Moreover, inter- and intratumor heterogeneity raise the demand for highly personalized, precision medicine to an increasing degree when targeting neoantigens. Most tumor mutations remain private and are specifically found within one patient. Therefore, the enormous tumor specificity of neoantigens turns them into safe immunotherapeutic targets, in theory. Since somatic mutations or posttranslational modifications lead to an aberrant immunopeptidome, neoantigens are predicted based on the whole exome and transcriptome sequencing of tumor tissue compared to normal body cells, as well as HLA-binding affinities [168,169]. Their actual identification by mass-spectrometric (MS) analyses, however, remains laborious and has limited sensitivity [172]. 

Furthermore, due to spatial and temporal genomic diversity between tumor clones within one genetically unstable tumor, the likelihood of tumor-wide clonal antigen expression is reduced. Detected by sampling from several regions of the same tumor, it has been reported that the local predominance of certain tumor subclones (with their individual surface immunopeptidomes) leads to the regional expansion of disparate reactive T cell clones. This is most likely occurs due to genetic determinants as well as in response to the surrounding TME [173,174,175]. This underlines the need for the heterogeneity assessment of tumors to detect important driver mutations and downstream broadly expressed and probably promising peptide targets. 

Neoantigens have been widely discussed in the context of immunotherapy over the past few years, since a high tumor mutational burden (TMB) correlates with better clinical outcomes upon immunotherapy, especially checkpoint inhibition [159,176]. At the same time, McGranahan et al. reported for non-small cell lung cancer (NSCLC) that it is not the pure quantity of neoantigens alone, but within a cohort of a high TMB the homogenous, clonal neoepitope expression, which is associated with prolonged survival. A highly heterogeneous distribution of neoantigen profiles correlated with deteriorated outcomes. [159] Subclonal mutations, occurring only in a fraction of the tumor and often selected during aggressive chemo- or radiotherapy regimens, may aggravate prognoses. Not only does this stress the need for the better identification of clonally and subclonally expressed neoepitopes that can predict the optimal antigen candidates for TCR-based ACT, but it also questions the use of immunotherapy for solely relapsed, refractory patients. By tracking the driver mutations shared between several patients, such as in PIK3CA, the first examples of public neoantigens and cognate TCRs have been identified. They carry implications for less personalized, off-the-shelf neoantigen-targeted ACT treatment possibilities in the future [177]. However, to date, the search for public neoantigens has been rather unsuccessful. 

##### HLA-Expression

Interestingly, the criteria of a high neoantigen load and intratumor homogeneity only predicted therapy outcomes for unknown-stage lung adenocarcinoma (LUAD), not for lung squamous cell carcinoma (LUSC), despite a similar neoantigen load in the studies of the group around McGranahan. Most likely, a lower expression of HLA-molecules on the tumor cell surface and aberrant antigen presentation mechanisms are responsible [159]. Immune escape by the downregulation of the antigen presentation machinery suggests a further major challenge for TCR-based ACT apart from the identification of suitable target structures; even if CTAs, self-antigens, or neoepitopes are present and identified in the cancer cell, they are not necessarily presented on the cell surface—mostly a result of a high immunologic selection pressure [178,179,180,181]. The irreversible structural, genomic loss of HLA in a tumor cell is frequently caused by the loss of heterozygosity (LOH) of chromosome 6p21, harboring HLA-ABC, or LOH in chromosome 15, carrying the β2-microglobulin (β2m) gene [182]. As they can only be corrected by replacing the defective gene, attempts were made to recover HLA-I-expression by adenoviral vectors in cell lines. However, the targeted expression in the tumor tissue of a patient could only be reached by the local administration of such a vector into the lesion [183]. 

Otherwise, these irreversible mutations inevitably required MHC-independent treatment. Either another immunogenic epitope, presented on a still-expressed HLA, must be identified, or CARs, natural killer (NK) cells, or chemo- or radiotherapeutic approaches may be more suitable to eliminating these tumor clones than TCRs. For example, the complete loss of HLA type I-molecules and thereby those inhibitory to NK-cells (missing self) renders tumor cells susceptible to NK-mediated killing [184]. However, a downregulation without a complete loss avoids NK- and T cell-mediated immune surveillance [185]. If abnormalities in antigen presentation remain reversible by cytokine administration, various stimuli might be able to upregulate HLA expression. It has been shown, for instance, that IFN-γ or TNF-α could upregulate HLA-expression and thereby contribute to more effective immunotherapy [186,187,188]. In principle, it must again be stressed that an inflammatory stimulus in the tumor tissue is necessary for increasing antigen presentation.

##### TCR Identification and Assessment

Tumor heterogeneity can be influenced only to a limited extent, and immune selection pressure almost necessarily favors tumor clones with defective antigen presentation. Thus, beyond attempts to engineer the TME, optimizing the living drug itself and its receptor is necessary to support TCR-T cell ACT.

First, receptor composition matters; even assuming the targeting of a clonal mutation present in all tumor cells that might be a driver of mutation in the tumor and therefore be responsible for the tumor phenotype, one single TCR will most likely not be able to durably eradicate the whole tumor without any escaping cancer clones. After successful ICI therapy, there is an at least oligoclonal TCR repertoire expected to be tumor-reactive in responding patients. Despite the difficulty in assessing their role in tumor debulking, their increased abundance upon successful ICI therapy renders the contribution of all expanded TCRs highly likely. [189] Therefore, it might be worth considering mimicking polyclonality in TCR-based ACT and infusing a combination of different TCRs instead of one alone. Bioinformatical methods could help to assess the clonality of antigen candidates [190] and immunopeptidomics or RNA-sequencing could validate peptides beyond prediction algorithms [172].

The identification of the respective TCRs for these mutations remains the second current bottleneck for such an ACT pipeline. Seeking to overcome the low sensitivity for the identification of small populations of tumor-specific TCRs within nonspecifically expanded bulk TIL populations (in some cases up to 11% or more) or peripheral blood mononuclear cells (PBMCs) (ranging between 0.002 and 0.4%) [191], single-cell sequencing approaches have recently been launched to lower the detection limit for tumor-reactive TCRs [63,64,65,192]. Transcriptomic signatures could be employed to identify tumor-reactive T cells amongst TILs, which can often be distinguished by effector and dysfunction markers, and therefore broaden the number of receptors identified [63,65].

##### Influence of T Cell Stimulation Strength on T Cell Function and Presence within the Tumor

Assuming that TCR identification were to be facilitated, it will still be crucial to understand which qualities of a TCR are associated with the most potent antitumor functions. The paradigm of ACT throughout the past few decades has focused on increasing the avidity of TCRs by affinity maturation procedures [193,194] since a high avidity is regarded as the major predictive factor for tumor eradication. A higher avidity is associated with higher activation levels, a stronger antitumor response, and more rapid cell lysis, even at lower doses of antigen [195,196,197,198]. Nonetheless, TCR clonotypes with relatively lower avidity levels are also able to expand in response to tumors and remain present at often unexpectedly high frequencies in the TCR repertoire of some patients [63,199,200]. In fact, excessively high levels of activation might overstimulate T cells and be associated with an impaired proliferative capacity as well as dysfunction. For example, MAPK-signaling was deteriorated by the overstimulation of T cells [201]. A “goldilocks” model for the optimal T cell activation for in vivo settings has been suggested [202,203] and, recently, an intermediate level of activation was described as capable of improving antitumor activity in vivo [204]. Shakiba et. al. showed that the counter-intuitive removal of the CD8-domain in T cells with a very strong initial level of activation, which thereby attenuated the signal strength in the T cells in their system, enabled targeted cell killing [204]. Therefore, this work challenged the current paradigms of avidity in ACT and asked the question: which engineering efforts of the TCR and the co-signaling receptor repertoire might be fruitful?

The large influence of the addition of costimulatory domains onto T cells became especially clear by CAR-T cell engineering [205]. In TCR-T cells to date, fusion receptors, for instance, of extracellular TIGIT- and intracellular CD28-domains, were tested for decreasing inhibitory signals [206], while CD40L and CD28 were tested for increasing the activation signals [207] on effector T cell populations. Such fusion receptors can be specifically designed for and adapted to the respective malignancies, responding to upregulated inhibitory receptors deteriorating T cell functionality. This was exemplarily demonstrated by attempts at fusing CD200R and CD28 domains for the treatment of AML, where CD200 (OX2) expression often dampens T cell responses [208]. CD19-directed CAR-like receptors with costimulatory signaling domains, but no CD3ζ-chains were employed in the B cell entities to provide costimulatory signals to the T cells, circumventing the lack of corresponding costimulatory ligands in most tumor entities [209]. However, studies such as the work of Shakiba et al. [204] illustrated that simply increasing the activation level upon TCR ligation might not necessarily strengthen functionality. Instead, all such engineering approaches must be tightly balanced within and adapted to the existent system, where numerous different factors influence T cell activation. These include TCR intrinsic properties, antigen density, HLA-expression, the amount of tumor cells, activatory and inhibitory co-signaling receptor–ligand interactions, and many other factors.

##### Administration of CD4-T Helper Cells

CD8^+^ T cells, naturally, were the focus of antitumor therapy due to their cytotoxic capacities [210]. However, the reinfusion of CD4^+^ T cell products was demonstrated to result in a partly impressive tumor regression, which pointed to the therapeutic potential of CD4+ T cells [211,212]. On the one hand, they strongly trigger CD8-responses; CD4+ T cells secrete IFN-γ upon antigen encounter, which in turn upregulates MHC-class I and II molecules in the TME and especially on antigen-presenting DCs. They also interact with APCs themselves through costimulatory interactions (e.g., CD40-L and CD40), activating the DCs to migrate towards lymph nodes and present tumor antigens to CD8^+^ T cells [213]. Moreover, cytotoxic CD4^+^ T cell states can directly mediate antitumor responses [214,215,216]; however, such responses are MHC-class II-dependent, which is not expressed in most solid tumor entities. The CD8-dependency of most MHC-class I-restricted TCRs complicates their use in CD4^+^-T cells [217]. Only a few clinical trials combining CD8^+^ and CD4^+^ T cells for adoptive TCR-T cell transfer have been launched to date (e.g., NCT04639245). For CAR-products used in the treatment of B cell lymphomas, an advantage of combination products could be shown [218,219]. Therefore, the T cells’ help from CD4^+^ T cells could support the eradication of the tumor and the recruitment of more CD8^+^ T cells by setting an inflammatory stimulus.

##### CAR-T Cells in Solid Tumors: Armored CARs

In contrast to TCR-based ACT, it was much easier for CAR approaches to test the above-mentioned combinations of CD4^+^ and CD8^+^ CAR-transgenic T cells, as both are able to target the same antigen MHC independently. However, the success in solid tumors was limited. It is assumed that, besides the lack of specific antigen, the cytokine-stimulation for T cell activation lacks CAR-T cell signaling in solid tumors. Several groups have started to equip a fourth generation of CAR constructs with additional features to increase their safety, specificity, and activation strength. These features include reversible and irreversible on-and-off switches, cytokine secretion elements, chemokine receptors, and more. These “weapons” of the armored CARs have been reviewed elsewhere in more detail [220].

One impressive attempt to ameliorate tumor recognition was the employment of a synthetically engineered form of the Notch protein, released from its membrane-anchor by intramembrane proteolysis upon the extracellular binding of its cognate ligand. By replacing the extracellular sensing domain, as well as the intracellularly released transcriptional module, this technology was used to create new cell–cell interaction circuits, allowing for even multiple different chimeric receptors per cell. The platform became known as Synthetic Notch (synNotch) technology [221]. By applying this technology to a defined set of several different antigens—which are not entirely tumor-specific on their own, but in combination—the controlled local expression of a CAR can be achieved and surface proteins that are not tumor-specific per se can be accessed, without the risk of systemic toxicity. Meanwhile, the constant tonic signaling in the T cells due to the overexpressed, constantly weakly stimulating CAR on its surface can be reduced and the cell persistence improved [222].

Despite all engineering already performed on CAR constructs, phosphoproteomic analyses of TCR- and CAR-T cells have highlighted the inability of CAR signaling to entirely mimic physiological TCR signaling cascades. Salter et. al. [223] described, by comparing ROR1-CAR and EBV-specific TCR-activated T cells, the lower phosphorylation of the canonical T cell signaling molecules CD3δ, CD3ε, and CD3γ as well as a linker for the activation of T cells (LAT) in CAR-cells. Consequently, they suggested the supraphysiological phosphorylation of CD3ζ and CD28 to overcome unphysiological T cell activation [223]. They illustrated the level of highly fine-tuned engineering needed to precisely intervene in signaling pathways and potentially improve CAR performance in solid tumors.

As the ultimate goal of a CAR-receptor is to simulate (or maybe even overtake) TCR signaling, other CAR-approaches have coopted the endogenous TCR of a cell and used the association of the natural CD3-complex with the TCR instead of CAR-intrinsic CD3-chains to activate the T cell downstream. These T cell antigen coupler (TAC)-receptors outperformed CD28-based CARs in a solid tumor model, exerting a more efficient tumor rejection, reduced toxicity, and increased tumor infiltration [224]. Thus, on the one hand, CARs demonstrated that T cells could be reprogrammed toward tumor rejection with a construction kit of different receptor and signaling domains that could be systematically exchanged and manipulated to optimize cellular activation. On the other hand, CAR-T cells also underlined the enormous complexity of circuits activated downstream from a TCR, which are neither fully understood nor easy to mimic in their entirety. It will be interesting to see whether synthetically engineered CARs or naturally encoded TCRs will dominate ACT in solid tumors in the future when the diversity of engineering strategies—some of which are outlined above—have been tested in clinical studies to assess the superiority of one approach over the other. In principle, all these manipulations launched in armored CAR-T cells could also be transferred to TCR-T cells, which could also shed more light on the role of the receptor itself for the initial activation, persistence, and dysfunction.

##### Receptor Specificity: On- and Off-Target Dose-Limiting Toxicity

Specificity for the tumor tissue remains another major bottleneck for ACT regarding dose-limiting toxicities. In the past, serious adverse events due to off-target toxicities of TCRs occurred, such as neurotoxicity after a previously disregarded MAGE-A12 coexpression in the brain [56] or a cardiac failure due to a cross-reaction with an unrelated (titin-derived) peptide in the heart [225]. These toxicities demonstrated the need for improved screening methods for the potential cross-reactivity of TCR, which cannot be covered by mouse models alone. Each small genetic variance between individuals can potentially create epitopes, resulting in off-target toxicity. However, on-target off-tumor toxicity must be considered when evaluating the safety of TCR-based therapies. For example, while targeting TAAs such as MART1, patients suffered from uveitis and hearing loss due to the destruction of normal melanocytes in the skin, eye, and cochlea [226], while severe inflammatory colitis was observed during a treatment with carcinoembryonic antigen (CEA)-specific TCRs [227].

The same risk for on-target off-tumor toxicity has been observed for CAR T cells. The infusion of an ERBB2-targeting CAR into a colon cancer patient, which led to immediate severe respiratory distress and lethal cytokine release syndrome (CRS), impressively demonstrates this. Dramatic pulmonary infiltrates suggested the recognition of low ERBB2-levels on the lung epithelium as causal [228]. Therefore, one major goal of engineering CAR-T cells is directed toward increasing safety through the inclusion of suicide domains into CAR-constructs, such as inducible caspase 9 (iC9). By applying agents that bind to the promotor regions of this gene, CAR-T cells are eliminated irreversibly from the patient; thus, the severe side effects should be reduced [229]. For a more elegant application, reversible control over CARs—on-and-off switch domains that temporarily inhibit CAR-signaling but do not destroy them entirely—have been transduced to CAR-T cells [230,231]. These approaches demonstrate the possibility of gaining more precise control over the doses and toxicities of ACT. Further engineering of CAR platforms aims at increasing safety, e.g., by the ligation of two different antigens in the target tissue or the introduction of an interposed soluble adaptor module providing antigen specificity and a universal binding motif recognized by CAR-T cells [232,233,234].

### 3.3. Overcoming T Cell Exhaustion and Dysfunction

#### 3.3.1. Hostile Environment for T Cells in the Tumor

As stated earlier, TIL therapy not only impressively demonstrates the reinvigoration of T cells towards antitumor activity but also poses the question of why these T cells could not control the tumor in the first place. Beyond infiltrating the tumor, their high frequencies in the TCR repertoire suggest that the TCR clones had most likely already recognized their target. Still, these T cells became immunosuppressed in the TME at a certain point. Several different immunosuppressive mediators (e.g., TGF-β, IL-10, prostaglandins, soluble FAS, adenosine, and reactive oxygen species (ROS)) and ligands were secreted or expressed in the TME by tumor cells, cancer-associated fibroblasts (CAFs), myeloid-derived suppressor cells (MDSCs), tumor-associated macrophages (TAMs), neutrophils, mast cells, or T_regs_ [235].

Of these, the latter may be directly involved in the inhibition of effector T cell functions via the inhibition of costimulatory signals of CD80 and CD86 on the DCs via CTLA4, competitive IL-2 consumption by high-affinity IL-2 receptors (CD25), the secretion of inhibitory cytokines such as TGF-β or IL-10, and further mechanisms [236]. T_reg_ in the TME express several surface markers, including CD25, CTLA4, PD-1, ICOS, GITR, CCR3, and CCR8. These markers can be used to deplete T_reg_ cells, which is primarily attempted by the use of antibodies [236]. For example, CAR NK cells targeting CD25 have been tested with the aim of depleting T_reg_ populations and overcoming immune escape [237]. However, since many of these surface receptors are shared with activated effector T cells, a more precise counteraction of T_reg_-associated immunosuppression is needed. By the CRISPR/Cas9-mediated knockout of the endogenous TGF-β receptor II (TGFBR2) in CAR T cells, their conversion towards T_reg_ phenotypes can be prevented while increasing the in vitro and in vivo effector function of these CAR T cells in TGF-β-rich TMEs [238]. As mentioned earlier, introducing a dominant-negative TGF-β receptor lacking the intracellular signaling motif, but directly competing with the wild-type receptor, could scavenge TGF-β and prevent its immunosuppressive activity [139,150].

Furthermore, hypoglycemic and hypoxic conditions in the tumor detrimentally affect T cell activation. Activated T cells switch from oxidative phosphorylation to glycolysis, facilitating their fast proliferation and fiercely competing with tumor cells for glucose. A lack of glucose results in lowered AKT/mTOR signaling, downregulated glucose receptor 1 (GLUT1)-expression, reduced glycolysis capacity, and lowered IFN-γ secretion [239]. Moreover, supported by the reduced uptake of glucose, mitochondrial fragmentation and the production of large amounts of ROS metabolic impairment in CD8^+^ TILs has been reported. This points to another important metabolic stressor for T cells in the TME: oxidative stress [240]. To increase the protection of infused CAR-T cells from ROS-inflicted oxidative stress, the coexpression of the hydrogen peroxide-reducing enzyme catalase, for example, leads to improved proliferative and cytotoxic profiles in CAR-T cells [241]. This implicates the potential metabolic switches might have for T cell engineering in the TME.

#### 3.3.2. T Cell Exhaustion due to Chronic Activation

The factors described above support a hostile environment for intratumoral T cells by counteracting their activity. However, constant TCR signaling itself might be one of the most important factors impairing T cell fitness. Throughout recent years, more attention has been paid to the investigation of dysfunctional T cell states in the TME. Transcriptomic signatures of tumor-reactive TCR clonotypes have been explored on a single-cell level and can be distinguished from virus-specific bystander T cell populations by the upregulation of several inhibitory receptors and an overall dysfunction-associated signature [63,65]. Comparing the TME-setting to acute infections, the persistence of the antigen instead of its clearance leads—similarly to chronic viral infections—to continuous TCR signaling, which eventually exhausts T cells. This perhaps most prominent form of T cell dysfunction in the tumor setting, termed T cell “exhaustion”, is generally defined by functional impairment associated with an upregulation of inhibitory surface receptors such as PD-1, Tim3, Lag3, or TIGIT [242,243]. Numerous investigations have categorized subgroups and precursor populations within the exhausted T cells [244,245]. Recently, a four-stage model suggested a transgression in a hierarchical order, dependent on the transcription factors TCF1, T-BET, and TOX, from the TCF1+ progenitor toward terminally exhausted cells [245]. TCF1+, non-exhausted, tumor-reactive T cell clonotypes were detected in the draining lymph nodes. Direct migration to the tumor tissue suggested one possible source for the replenishment of intra-tumoral T cell populations and posed the question of whether the non-exhausted precursor population was situated in the tumor or in the surrounding lymph nodes [246]. To date, it remains elusive which T cell population in the TME exerts antitumor activity most potently, which one replenishes the pool of tumor-reactive T cells, and whether those two are identical. Determining both subsets will be crucial to improving engineering strategies and tilting the balance in the TME back to T cell functionality.

#### 3.3.3. T Cell Senescence

Dysfunction is not solely caused by exhaustion. It also appears in the context of senescence. Since exhausted and senescent cells share several overlapping phenotypic and functional features, such as a defective effector capacity, cell cycle arrest, and impaired proliferation, it is hard to distinguish them from each other. In contrast to exhaustion, senescence depends more on the replicative shortening of telomeres in the cell (replicative senescence) or other external disruptive factors, resulting in cellular damage (e.g., DNA damage, oncogenes, oxidative stress, chemotherapy, and mitochondrial dysfunction). Yet, the reasons for senescence are not fully understood [247,248]. Not surprisingly, the injection of initially more stem-like, less differentiated and as such naïve stem cell memory or central memory T cell populations (T_N_, T_SCM_ or T_CM_) has been associated with improved therapy outcomes due to increased T cell fitness [249,250]. Instead of enriching these populations, other approaches try to retard T cell differentiation through the addition of PI3Kδ-inhibitors during T cell culture, which at least partly restores functional capacity [251].

#### 3.3.4. Strategies to Counteract T Cell Dysfunction

Since antigen encounter is necessarily linked to cellular differentiation, completely preventing all forms of dysfunction will not be feasible. However, both forms of dysfunction in T cells are reversible up to a certain point, which bears therapeutic potential through the reversion of T cell fates in the TME [252]. First, the intrinsic regulators of exhaustion and senescence can be targeted to reverse cellular inhibition. The interactions of inhibitory receptors upregulated during chronic TCR stimulation and their ligands in the TME become known as immune checkpoints and various treatment regimens based on ICI are already in the clinical stage. Programmed death receptor-1 (PD-1), which is transiently upregulated during T cell activation [253], is the most illustrative example of an inhibitory receptor that is consistently and highly expressed in exhausted T cells. Its primary biological function is the maintenance of T cell responses within a desired physiological range, preventing autoimmunity. According to its expression pattern, PD-1 primarily depicts a marker of activated T cells of which exhausted and thereby chronically activated cells represent one group [254]. Upon binding to its ligands, namely, PD-L1 or the lower expressed PD-L2, several intracellular mechanisms, which are not understood in detail, dampen T cell effector functions. The involvement of tyrosine-protein phosphatases SHP1 and SHP2 as well as a role of the phosphoinositide 3-kinase (PI3K), AKT, and RAS pathways were reported, amongst other effects on T cell signaling [255,256]. Preventing this receptor–ligand interaction by an administration of either PD-1- (pembrolizumab and nivolumab) or PD-L1- (avelumab or atezolizumab) blocking antibodies could achieve a convincing prolongation of progression-free survival (PFS) in multiple clinical trials across different entities [2,3,257,258,259,260].

However, monotherapy with PD-1 blockade alone generally does not lead to durable remissions in most malignancies. In advanced melanoma, for example, a combination therapy of nivolumab and ipilimumab performed better. [261,262] The latter is a blocking antibody for cytotoxic T-lymphocyte-associated protein 4 (CTLA-4), dampening T cell activation by competition with the costimulatory molecule CD28 for its ligands (B7-1/CD80 and B7-2/CD86) and thereby depriving T cells of important costimulatory signals [263]. This superiority of combination therapy is not surprising, considering the compensatory upregulation of other non-redundant inhibitory receptors upon the blockade of one single inhibitory pathway [264,265].

Since it has been suggested that ICI therapies unleash T cell activity within the preexistent tumor-specific repertoire, it was reasonable to test those agents in combination with adoptively transferred T cells. Preclinical studies combining CAR-T cells and PD-1 blockade resulted, in some studies, in enhanced proliferation and effector functions (e.g., IFN-γ production and granzyme B expression) [266,267], while pembrolizumab reinvigorated already exhausted CAR-T cells [268]. Nonetheless, preclinical data substantiating an effect of anti-PD-1 antibodies on adoptively transferred T cells are currently lacking. Several clinical studies for testing the combination of systemically administrated checkpoint inhibitors and CAR-T cell therapies have been launched to date (see Table 2). While some of these trials yielded promising responses, a PD-1 blockade could not further enhance T cell expansion or tumor regression in others [269,270]. Since the systemic administration of checkpoint inhibitors comes at the expense of several toxicities [271,272], other forms of administration might be considered. The local production of PD-L1 antibodies delivered by an oncolytic adenovirus in the TME enhanced CAR-T cell killing [273]. Further combining this approach with a CD44v6-specific BiTE and the immunostimulatory IL-12—all expressed by the same virus—significantly improved tumor control, mediated by HER2-specific CAR T-cells and non-transgenic native TCRs engaged via the BiTE included [274]. This impressive number of combined therapeutic approaches underlines the relevance of combinatorial approaches, owing to the fact that no single cancer immunotherapy will likely defeat all the evasion mechanisms of solid tumors on its own.

Instead of a systemic or local administration of blocking antibodies, T cell-intrinsic engineering strategies could also aim at unleashing antitumor effector functions. Some examples for fusion receptors of TIGIT or CD40L and CD28 [206,207] have been outlined above. Beyond manipulating single extracellular receptors, various preclinical models have investigated the manipulation of whole transcription factor axes for ACT. Mackall and colleagues forced the overexpression of c-Jun in CAR-T cells, leading to exhaustion resistance with reduced PD-1- and CD39-expression, as well as enhanced functional capacities by the direct activation of AP-1 [276]. In another example, TOX- and NR4A-deletion—both involved in a positive feedback loop inducing exhaustion in PD-1^high^ TIM-3^high^ CAR-T cells—increased cytokine secretion and decreased inhibitory receptor expression. [277] The various pathways of T cell signaling potentially useful as adjustment screws in improving ACT approaches, have been reviewed elsewhere in more detail [278].

The first pilot trial of CRISPR-Cas9-engineered NY-ESO-1-TCR-transgenic T cells infused into patients within a clinical study was recently published by Stadtmauer et. al. [279]. On top of deleting the α- and β-chain locus of the endogenous TCR of these T cells to reduce the mispairing of TCR chains, PD-1 was knocked out (KO) in these T cells. In previous publications, PD-1-deficient CAR-T cells were shown to exert a superior antitumor function in xenograft mouse models [280,281]. Yet, in this study, the contraction of putative PD-1 KO-T cells identified by sequenced editing events in the gene from 25% to 5% of all T cells within four months after an adoptive T cell transfer rather suggests a deficit in longevity [279]. Other publications assigned a critical role to PD-1 in limiting the early overactivation and promoting the stability of exhausted T cell responses to chronic antigen exposure [282]. Inhibitory receptors, such as PD-1 or LAG-3, upregulated on T cells upon activation—transiently during early activation and permanently in exhausted cells—may be essential as a rheostat function for T cells [283,284].

So far, most engineering strategies in adoptive T cell transfer aimed at an increase in the activation potential and stimulation to in turn increase antitumor response. Still, it was reported that the exhaustion and dysfunction in T cells was determined by the initial T cell activation early after antigen encounter [285]. As mentioned earlier, Shakiba et. al. recently showed that the TCR stimulation strength was responsible for the maintenance of T cell effector function and that overly high and overly low stimuli impaired functionality [204]. The group suggested an intermediate stimulation level as optimal for the preservation of T cell populations and questioned the design of T cell engineering strategies, which could be highly dependent on the individual TCR employed for ACT in its interplay with the tumor cells [own unpublished data]. An increase in T cell activation might be beneficial for some TCRs (or CARs) as well as for initial effector function and tumor rejection, which require strong T cell activation. On the other hand, this could come at the expense of impairing T cell longevity. The long-term maintenance of T cell populations and durable tumor-control in the patient were still highly limited throughout the clinical studies for ACT, even amongst the most immunogenic entities [286]. It is possible that both require a much finer, more precise modulation of T cells (e.g., in TCR strength, the interplay of activatory and inhibitory cosignaling molecules) with respect to antigen density and co-signaling molecules expressed by the tumor. Modifications of therapy regimens (e.g., multiple infusion timepoints with differently modified T cells) may help sustain T cell functionality.

## 4. Conclusions

Despite the substantial progress in ACT over the past several years, solid tumors remain a major challenge. At best, the current technologies help to debulk the tumor burden, but permanent complete remissions are still rarely observed. Most likely, a single receptor—TCR or CAR—or even several will not be able to eradicate an entire tumor without immune escape. Therefore, setting an inflammatory trigger in the TME that results in the reinvigoration of endogenous adaptive and innate immune responses will be an important aim for T cell-based ACT in solid tumors. So far, we have not found a “master solution” for permanently tilting the equilibrium in the TME back to tumor control, and it is probable that no single therapeutic regimen will fit every patient. However, the multitude of engineering strategies for TME, as well as transferred T cells themselves, of which some were outlined in this review, show great potential for combinatorial treatment strategies. Tedious strategic preclinical and clinical assessments of all these tools will be necessary to find synergisms and will gradually pave the way for transgenic T cells into solid tumors.

## Figures and Tables

**Figure 1 cancers-14-04192-f001:**
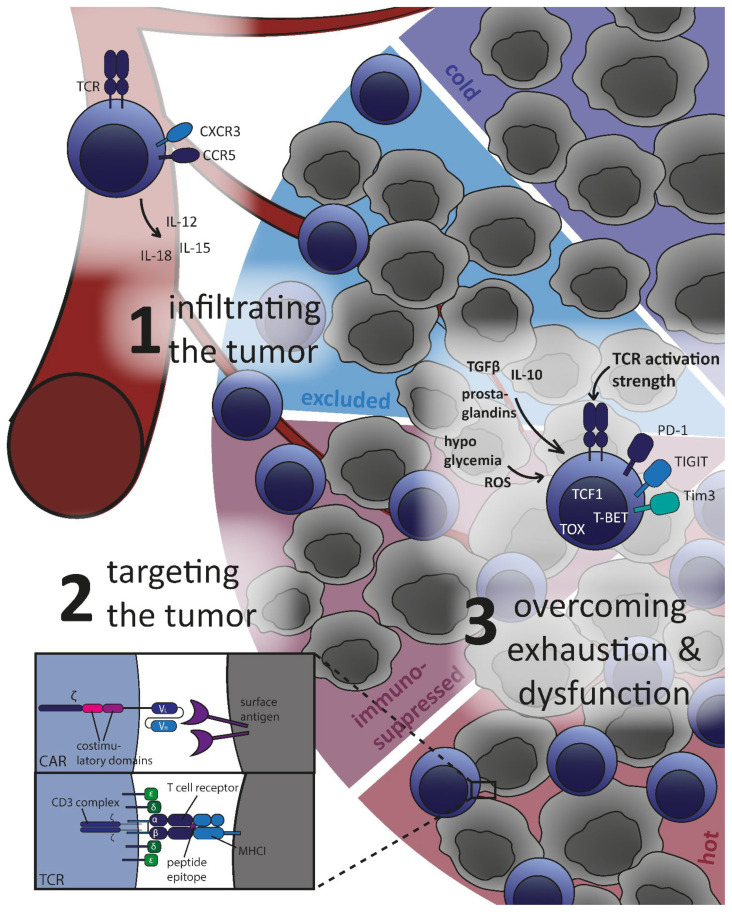
Major challenges for adoptively transferred T cells in the tumor microenvironment. The immune infiltration of a solid tumor is schematically represented by a cold (dark blue), an excluded (light blue), an immunosuppressed (violet), and a hot (red) tumor sector with gradually more CD8+ T cells (shown as blue cells) penetrating the tumor tissue (represented by grey tumor cells). Three major challenges are depicted, along with relevant exemplary adjustment screws, as discussed in this review: infiltrating the tumor (1), targeting the tumor via CARs or TCRs (both illustrated schematically) (2), and overcoming exhaustion and dysfunction (3).

**Table 1 cancers-14-04192-t001:** Selected clinical studies for TCR-T cell-based therapies in solid tumors (information from clinicaltrials.gov (accessed on 6 June 2022)).

Target	Target	TCR	HLA	Entities	Sponsor	Phase	n ^1^	Start	Further Therapy	Study ID ^2^	Ref.
TAA	WT-1	WT1-TCRc4	HLA-A*0201	mesothelioma, NSCLC (both stage III-IV)	Fred Hutchinson Cancer Center/ Juno Therapeutics	I/II	11	2015	Aldesleukin	NCT02408016	
	MSLN	FH-TCR-Tᴍsʟɴ	HLA-A*0201	pancreatic ductal adenocarcinoma	Fred Hutchinson Cancer Center/ Juno Therapeutics	I/II	15	2021		NCT04809766	
CTA/ Oncofetal proteins	NY-ESO-1	Anti-NY ESO-1 mTCR PBL	HLA-A*0201	melanoma, meningioma, breast CA, NSCLC, HCC	NCI	II	11	2013	Aldesleukin	NCT01967823	
	NY-ESO-1	Anti-NY ESO-1 mTCR PBL	HLA-A*0201	melanoma, renal cell cancer, metastatic cancer	NCI	II	45	2008	Aldesleukin	NCT00670748	[55]
	NY-ESO-1	NY-ESO-1c259T	HLA-A*0201, HLA-A*0205, and/or HLA-A*0206	melanoma	Adapt immune/ Glaxo SmithKline	I/II	4	2011		NCT01350401	
	NY-ESO-1	NY-ESO-1c259T	HLA-A*0201, HLA-A*0205, and/or HLA-A*0206	ovarian cancer	Adapt immune/ Glaxo SmithKline	I/II	9	2013		NCT01892293	
	NY-ESO-1	NY-ESO-1c259T/GSK3377794	HLA-A*0201, HLA-A*0205, and/or HLA-A*0206	liposarcoma	Adapt immune/ Glaxo SmithKline	II	23	2016		NCT02992743	[67]
	NY-ESO-1	TBI-1301	HLA-A*02:01 or HLA-A*02:06	sarcoma, melanoma, esophageal, ovarian, lung, bladder, or liver cancer	University Health Network, Toronto	I	22	2016		NCT02869217	
	MAGE-A3/12	PG13-MAGE-A3 TCR9W11	HLA-A*0201	metastatic cancer, metastatic renal cancer, metastatic melanoma	NCI	I/II	9	2010	Aldesleukin	NCT01273181	[56]
	MAGE-A3/12	Anti-MAGE-A3-DP4 TCR	HLA-DPB1*0401	melanoma, cervical, renal, urothelial, or breast cancer	NCI	I/II	21	2014	Aldesleukin	NCT02111850	[68]
	MAGE-A4	TBI-1201	HLA-A*24:02	various entities	Mie University	I	18	2014		NCT02096614	
	MAGE-A4	MAGE-A4^c1032^T	HLA-A*02	bladder, head and neck, ovarian, esophageal, gastric cancer, melanoma, NSCLC, synovial sarcoma, liposarcoma	Adapt immune/ Glaxo SmithKline	I	54	2017		NCT03132922	[69]
	MAGE-A10	MAGE A10^c796^T	HLA-A*0201 and/or HLA-A*0206	NSCLC	Adapt immune/ Glaxo SmithKline	I	28	2015		NCT02592577	[70]
	MAGE-A3/A6	KITE-718	HLA-DPB1*0401	various entities	Kite Pharma	I	16	2017		NCT03139370	[71]
	MAGE- A4/A8	ACTengine IMA201-101	HLA-A*0201	various entities	Immatics	I	22	2018		NCT03247309	[72]
	MAGE- A1	ACTengine IMA202-101	N.A.	various entities	Immatics	I	15	2019		NCT03441100	
	PRAME	IMA203-101 ACTengine	HLA-A*0201	various entities	Immatics	I	42	2019	IL-2, Nivolumab (Cohort B)	NCT03686124	[73]
	AFP	AFP^c332^T	HLA-A*02	HCC	Adapt immune/ Glaxo SmithKline	I	45	2017		NCT03132792	
neoantigens	personalized	NEO-PTC-01	Persona lized	melanoma	BioNTech	I	52	2020		NCT04625205	
		NeoTCR-P1		solid tumors	PACT Pharma, Inc.	Ia/Ib	148	2019	Aldesleukin, Nivolumab	NCT03970382	[74]
		N.A.		(neuro)endocrine tumors, NSCLC, ovarian, breast, GI cancers	NCI	II	270	2018	Aldesleukin, Pembrolizumab	NCT03412877	
		N.A.		malignant epithelial neoplasms	Providence Health & Services	I/Ib	24	2022	CDX-1140 (CD40 activation), Pembrolizumab	NCT04520711	

^1^ estimated number of patients enrolled or to be enrolled at the time of publication of this review, ^2^ source: clinicaltrials.gov.

**Table 2 cancers-14-04192-t002:** Selected clinical studies for combinatorial treatment of solid tumors: CAR-T cells and immune checkpoint inhibition (information from clinicaltrials.gov).

CAR	Target	CAR- Elements	Checkpoint Inhibitor	Entity	Sponsor	Phase	n ^1^	Start	Study ID ^2^	Ref.
CART-EGFRvIII	EGFR	41BB	Pembrolizumab	glioblastoma	University of Pennsylvania	I	7	2019	NCT03726515	
iC9.GD2-CAR3	GD-2	CD28 OX40 + iCaspase9	Pembrolizumab	neuroblastoma	Baylor College of Medicine	I	11	2013	NCT01822652	[270]
HER2-CAR T	HER2	CD28	Pembrolizumab/Nivolumab	sarcoma	Baylor College of Medicine	I	25	2021	NCT0499500/HEROS 3.0	
iCasp9M28z	MSLN	CD28 + iCaspase9	Pembrolizumab	malignant pleural disease, mesothelioma, lung Cancer, breast Cancer	Memorial Sloan Kettering Cancer Center	I/II	113	2015	NCT02414269	[275]
IL13Ra2-CAR	IL13Rα2	41BB + truncated CD19	Nivolumab + Ipilimumab	glioblastoma	City of Hope Medical Center	I	60	2019	NCT04003649	

^1^ estimated number of patients enrolled or to be enrolled at the time of publication of this review, ^2^ source: clinicaltrials.gov.

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
