# Peer review of "Paving the Way to Solid Tumors: Challenges and Strategies for Adoptively Transferred Transgenic T Cells in the Tumor Microenvironment"

_cancers, 2022, doi:10.3390/cancers14174192_

Round 1

Reviewer 1 Report

The review is interesting and deals with most of the issues concerning the use of CAR-T cells to treat solid tumors. However, in my opinion, the issue concerning TME infiltrating immune T cells should be addressed more extensively.

Some tips are provided to improve readability.

1) I would add a short introduction

2) the first paragraph discusses much more than T cell roles. I advise dividing the paragraph into 2-3 paragraphs

3) first paragraph. Lines 32-40. The sentence is not clear. Punctuation changes are required.

4) the second paragraph discusses some different points. I advise dividing it into at least 2 paragraphs

5) line 266. What does it mean "vascular normalization"?

6) line 605. Please delete the question mark

Other suggestions.

1) Line 52: CD25+FoxP3+ cells (i.e. thymus-derived Tregs) are not the Treg population most represented in TME. Peripherally-derived Tregs (among which Tr1, Th3, and GITR+ Treg) play a crucial role in TME and should be mentioned (see, for example, 10.1007/978-3-662-43492-5_3, 10.4049/jimmunol.178.1.179 , 10.1080/2162402X.2017.1387705)

2) Lines 250-264. The reasoning is not clear. The synergism anti-VEGF/anti-PD-1 could depend on the anti-angiogenic effect/T cell activation more than T cell recruitment. Please clarify.

3) line 265-... Another way to increase infiltration may be BCG treatment of bladder cancer

4) line 603-612. In my opinion, the exhaustion problem is relevant. In this context, some more escape strategies may be reported. For example, the studies by Cadilha et al (10.1126/sciadv.abi5781) and by Tang et al (10.1172/jci.insight.133977). Another crucial point for T cell exhaustion in TME is the role of tumor-infiltrating Tregs. In my opinion, it appears interesting the study by Dehbashi et al (10.3389/fonc.2021.649710).

Reviewer 2 Report

This Review deals with the effects of genetically modifies T cells and their interactions with the tumor microenvironment. The review is well organized and documented; I have two major comments

1. sentences like “Bioinformatical methods should be employed …”, “more effort should be put into finding a solution…” are clearly the opinion of the Authors . In the opinion of this Reviewer, articles like this should report only facts, not the opinions of the authors.

2. English language in understandable, but it is clearly not written by english-speaking people. Too many commas, too many expressions that are literally translated in English. It needs a profound revision.

Reviewer 3 Report

This is a comprehensive review of current transgenic T-cell therapy, successes and challenges. However, this manuscript requires significant editing as follows:

1)The length can be substantially reduced when excess words are removed.

2) The English usage, especially of non-scientific terms, is incorrect and makes the sentences have the wrong meaning as well as making the review difficult to read. This needs to be addressed. This is especially true when subjective terms are used. The use of commas throughout tends to be in the wrong place again making sentences hard to read. Some examples are given below (these are just examples, this occurs in nearly every paragraph):

Lines 25/26: this sentence makes no sense. Remove everything from the -both....

Line 41 remove "surely"

Line 49 change However to The

Sentence line 410-411 is a good example: Replace with  "Thus far the search for public neoantigens has been unsuccessful" 

As noted the word count can be cut and losing some of the non-scientific terms will also aid the readability of this document as the examples given above occur throughout.